# GigaVideo-1: Advancing Video Generation via Automatic Feedback with 4 GPU-Hours Fine-Tuning

## Abstract

Recent advances in diffusion models have significantly improved the quality of video generation. However, their real-world deployment often requires fine-tuning for physical constraints, a process that depends on human annotations and large-scale computational resources. In this paper, we propose *GigaVideo-1*, an efficient fine-tuning framework that advances video generation without additional human supervision. Rather than injecting large volumes of high-quality data from external sources, GigaVideo-1 unlocks the latent potential of pre-trained video diffusion models through automatic feedback. GigaVideo-1 focuses on two key aspects: data and optimization. On the data side, we design a prompt-driven data engine that constructs diverse, weakness-oriented training samples. On the optimization side, we introduce a reward-guided training strategy, which adaptively weights samples using feedback from pre-trained vision-language models with a realism constraint. GigaVideo-1 offers a flexible optimization framework adaptable to various capability dimensions. To demonstrate its versatility, we instantiate the framework on VBench-2.0's 17 evaluation dimensions as concrete application instances. Using Wan2.1 as the baseline, GigaVideo-1 yields consistent improvements, with an average gain of ∼4% using only 4 GPU-hours. Requiring no manual annotations and minimal real data, GigaVideo-1 shows both effectiveness and efficiency. Code, model, and data will be publicly available.

## 1 Introduction

Text-to-video (T2V) diffusion models have recently made notable progress in perceptual realism and stylistic diversity, synthesizing high-quality videos from natural-language prompts at scale (Gupta et al., 2024; He et al., 2024a; Wang et al., 2023a;b; Yang et al., 2024c). However, despite strong performance in surface-level attributes such as per-frame aesthetics and temporal smoothness, they often struggle with the deeper semantic understanding required for true realism. Deficiencies in these key dimensions continue to limit the real-world utility of generated content.

To mitigate these limitations, recent work has introduced fine-tuning pipelines aimed at improving generation quality along specific dimensions. These approaches generally fall into two categories. The first follows a supervised fine-tuning (SFT) paradigm, such as in Wan2.1 (Wan et al., 2025) and HunyuanVideo (Kong et al., 2024), where large-scale, high-quality datasets are used to continue training after pre-training. While effective, they demand extensive curated data and high computational cost. The second category, exemplified by methods like VideoAlign (Liu et al., 2025) and LiFT (Wang et al., 2024b), adopts human-in-the-loop reinforcement learning (RL) (Liu et al., 2024; Xu et al., 2024) to align with human preferences. These methods require manually annotated data with preference scores and the training of reward models, making them difficult to scale and resource-intensive.

To address these challenges, we introduce *GigaVideo-1*, a lightweight and data-efficient fine-tuning framework that enhances pre-trained video diffusion models without human supervision. Unlike methods that rely on large-scale external data injection, our approach activates the latent potential of foundation models through automated feedback mechanisms. This enables the model to self-improve based on its own behavior, dramatically reducing the need for curated datasets.

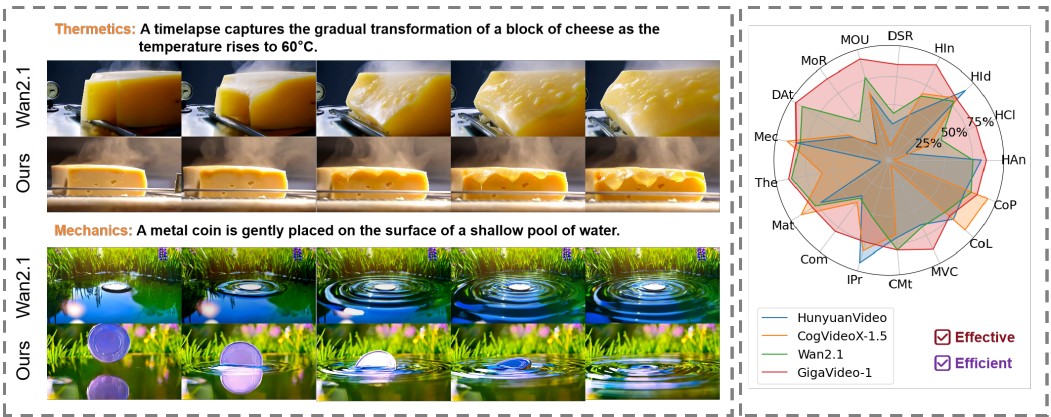

Figure 1: **Visualization of GigaVideo-1 performance**. The left figure compares videos generated by the baseline Wan2.1 (Wan et al., 2025) and our GigaVideo-1 across two different dimensions. The right figure provides the performance of GigaVideo-1 and other state-of-the-art T2V models on VBench-2.0. With only ∼4 GPU-hours of training, GigaVideo-1 achieves notable improvements over the baseline, demonstrating both effectiveness and efficiency.

Our method targets two key aspects of the fine-tuning process: training data and optimization strategy. To improve data quality, we propose a prompt-driven data engine that generates diverse training samples specifically oriented toward challenging or underrepresented generation attributes. For optimization, we introduce a reward-guided training strategy. This method leverages feedback from frozen vision-language models to adaptively weight training samples and applies regularization to maintain alignment with real-world data distributions.

Together, these components allow GigaVideo-1 to improve model performance across critical dimensions with only a small number of synthetic samples and minimal computation. Using 4 GPU-hours, our approach demonstrates that high-quality fine-tuning of T2V models can be achieved efficiently. Without human annotation or large-scale external data, GigaVideo-1 marks a promising step toward scalable, automatic refinement for video generation.

Our contribution can be summarized as follows:

- We propose *GigaVideo-1*, a lightweight fine-tuning pipeline for video generation. It improves pre-trained video diffusion models using automatic feedback without human supervision or large-scale data.

- From the aspect of fine-tuning data, we propose a prompt-driven data engine. It constructs diverse training samples by generating and expanding prompts specifically oriented toward model weaknesses.

- From the aspect of fine-tuning optimization, we propose a reward-guided training strategy. It adaptively weights training samples using feedback from pre-trained vision-language models and maintains distributional alignment with real data.

- Extensive experiments demonstrate the effectiveness and efficiency of our proposed method. Using minimum computation cost and external data, GigaVideo-1 realizes ˜4% performance gain over the baseline Wan2.1 (Wan et al., 2025), superior in almost all dimensions of the VBench-2.0 benchmark.

## 2 RELATED WORK

### 2.1 TEXT-TO-VIDEO GENERATION MODELS

Video generation (Chen et al., 2023; Harvey et al., 2022),(Yuan et al., 2025), particularly diffusion-based models (Ma et al., 2024; Zheng et al., 2024; Mei & Patel, 2023), has witnessed rapid progress in recent years. Among them, text-to-video (Singer et al., 2022; Zhang et al., 2024a; Zhou et al., 2024) generation has found broad applications in personalized content creation, enabling controllable

video synthesis based on user-provided prompts. A growing amount of research has demonstrated that scaling up both the data volume and model size leads to improved performance. As a result, numerous large-scale models have emerged, achieving strong results on general video generation tasks but often requiring extensive training data and computational resources. On the model structure side, researchers have also explored various innovations to improve model expressiveness and efficiency. For example, some frameworks replace the standard U-Net in diffusion models with Diffusion Transformer (Peebles & Xie, 2023) to enhance representation capacity. Some models, like Hunyuan (Kong et al., 2024), introduce 3D-VAE to better capture spatiotemporal dynamics. Another group of approaches (like SVD (Blattmann et al., 2023) and Wan2.1 (Wan et al., 2025)) uses improved modeling of the diffusion process, applying EDM (Karras et al., 2022) and flow-matching techniques (Esser et al., 2024) to accelerate diffusion inference and improve modeling efficiency, respectively. While these models perform well on general prompts, their outputs often fail to exhibit reasonable behavior in more structured or physics-intensive scenarios. For example, generated videos may violate basic physical principles such as object permanence or causality. New benchmarks such as VBench-2.0 (Zheng et al., 2025), WorldModelBench (Li et al., 2025) evaluate the effectiveness of the existing models in the above aspects. The results show that there is significant room for improvement in the behavior of these dimensions.

## 2.2 VIDEO GENERATION FINE-TUNING

Fine-tuning has emerged as an essential step for improving specific aspects of video generation, such as aesthetics and text alignment. Current methods can be broadly categorized into SFT and RL-based alignment, each with distinct advantages and limitations. SFT methods, such as Wan2.1 (Wan et al., 2025) and HunyuanVideo (Kong et al., 2024), extend model training using large-scale, high-quality datasets. These methods are effective at improving general-purpose generation, but they require significant computational resources and manual data curation, making them costly and inefficient.

The other fine-tuning strategy is reinforcement learning-based, aiming to align generation with human preferences using feedback-driven supervision. Among them, reward-weighted regression (RWR) (Peng et al., 1910; Lee et al., 2023; Furuta et al., 2024; Liu et al., 2025; He et al., 2022) assigns scalar rewards to generated samples and reweights their training loss. Although computationally efficient, methods like LiFT (Wang et al., 2024b) and FlowRWR (Liu et al., 2025) still rely on human-annotated data to train reward models. Another widely adopted class, direct preference optimization (DPO) (Rafailov et al., 2023; Wallace et al., 2024; Dong et al., 2023; Yang et al., 2024b; Liang et al., 2024), learns from pairwise human preference comparisons or ranked samples to guide model adaptation. Representative works include FlowDPO (Deng & Mineiro, 2024) and GAPO (Gu et al., 2025). Despite improved alignment with subjective quality, these methods are also constrained by the need for curated human preference data. Other RL methods, such as PPO (Schulman et al., 2017; Black et al., 2023; Fan et al., 2023) and online backpropagation (Yuan et al., 2024), are less common due to their high sampling cost and instability in high-dimensional video spaces.

While the above existing fine-tuning methods have demonstrated impressive improvements in generation quality, they either depend on massive training data with huge computation cost, or strongly rely on costly human-in-the-loop annotations. Meanwhile, these methods primarily focus on aligning with human preferences, rather than aligning with fundamental physical laws and commonsense principles in critical dimensions.

## 3 METHOD

### 3.1 FRAMEWORK OVERVIEW

We propose a lightweight and automatic fine-tuning pipeline for video generation, designed to enhance model performance efficiently using minimal data and compute. The design of our pipeline is motivated by two key factors governing fine-tuning efficacy in standard video generation frameworks (see Appendix A.1): the composition of training samples and the optimization strategy.

Accordingly, we introduce two core innovations, as illustrated in Fig. 2: a prompt-driven data construction mechanism and a reward-guided optimization strategy. Sec. 3.2 presents a data engine that diversifies prompt coverage and calibrates data distribution via synthetic video generation. Sec. 3.3 describes a reward-guided method that uses automated feedback to weight training samples

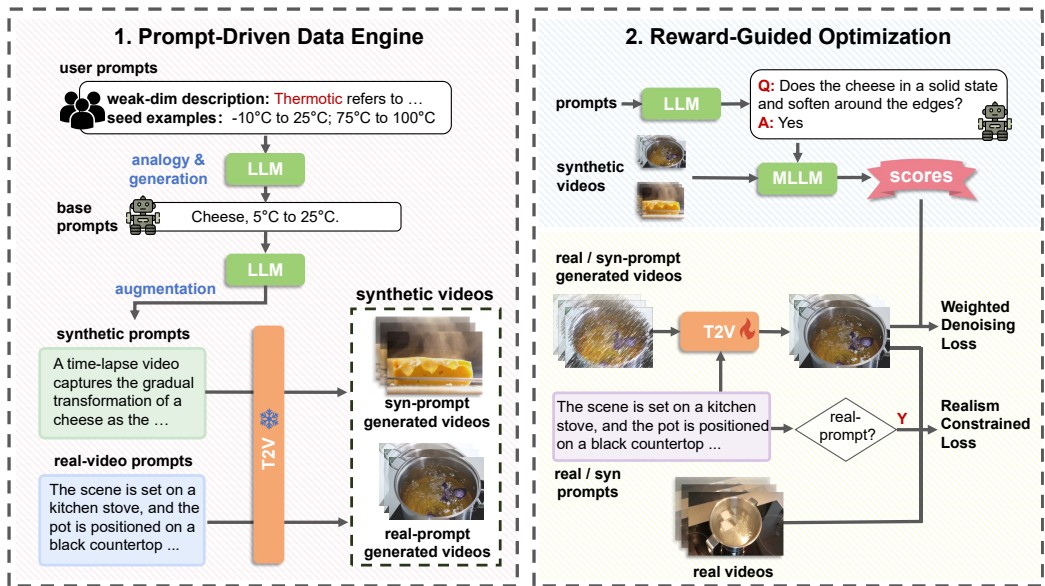

Figure 2: **GigaVideo-1 Training Pipeline.** Our pipeline consists of two components: prompt-driven data engine and reward-guided optimization. On the left, we generate synthetic prompts targeting weak dimensions using LLMs, and synthesize training videos via a pre-trained T2V model. These are combined with real-caption–based samples to balance diversity and realism. On the right, a frozen MLLM scores each video on dimension-specific criteria. These scores guide training via weighted denoising loss. For synthetic videos from real-world caption prompts, an extra realism constraint is applied for distribution alignment. GigaVideo-1 enables efficient, automatic fine-tuning without manual labels or extra data collection.

adaptively while maintaining realism. Together, these components form a modular and efficient framework for fine-tuning pre-trained video generation models without human annotations.

## 3.2 PROMPT-DRIVEN DATA ENGINE

Traditional video generation methods typically rely on real-world video-caption pairs to supervise training. While effective for general-purpose modeling, such data often reflects the natural distribution of events, scenes, and interactions. Elements like background clutter, occlusions, and multi-object dynamics are common. These factors dilute supervision when the goal is to improve targeted weaknesses, such as unrealistic motion or poor physical consistency.

To address this challenge, we introduce a prompt-driven data engine designed to construct targeted and controlled training samples. Rather than relying on captions extracted from real videos, we aim to generate synthetic prompts that directly highlight the dimensions where pre-trained models perform poorly. This allows us to bypass the uncontrolled variability of real-world content and focus learning on targeted failure modes.

We start by identifying some failure modes in the pre-trained model (e.g., unrealistic motion, implausible interactions) and selecting a few representative prompt examples that expose these issues. Each weak dimension is defined with a short description and a handful of seed phrases. These are then fed into a large language model using few-shot prompting to generate a diverse set of base prompts that are concise, visually grounded, and explicitly focused on the target concept (e.g., "Camera zoom in, Forbidden City"). More details and examples of the constructed base prompts can be found in the supplementary material.

We further expand these base prompts into stylistic variants by modifying scene elements such as lighting, motion type, or character appearance. This augmentation increases data diversity without deviating from the intended training objective. These prompts are then used to condition a pre-trained text-to-video model to generate synthetic videos. Compared to real-world footage, these synthetic samples are more focused and better aligned with the target dimensions.

This pipeline enables controlled data construction and effective fine-tuning without reliance on large-scale external human annotations. By eliminating irrelevant factors present in real-world data and amplifying targeted supervision signals, our method improves training efficiency and accelerates model adaptation on weak dimensions, even under limited computational budgets.

## 3.3 REWARD-GUIDED OPTIMIZATION

**VLM-based Feedback.** In conventional video generation training, the weighting coefficient $\lambda$ is typically set as a fixed constant to control the scale of the loss. However, this static setting fails to reflect the varying quality and usefulness of individual training samples, particularly when focusing on improving specific aspects of generation.

To address this, we propose a reward-guided optimization that leverages automatic feedback from pre-trained vision-language models. Instead of the human-supervised reward training like LiFT, we leverage multimodal large language model (MLLM)-based reward models as a strategic alternative. It eliminates the need for human-annotated data, enabling efficient large-scale evaluation at significantly reduced cost. Meanwhile, MLLM-based rewards excel in assessing physically-grounded dimensions by leveraging embedded scientific knowledge, often outperforming human annotations in consistency and accuracy. More analysis of the advantages in comparison is discussed in the Appendix F.

**Dimension-Specific Scoring.** To evaluate synthesized video–prompt pairs in a dimension-specific and interpretable manner, we leverage a large language model (LLM) to transform abstract quality dimensions into concrete, checkable criteria grounded in world knowledge. For each video, we first provide its associated prompt along with a target evaluation dimension (e.g., motion consistency or physical plausibility) to the LLM, which then generates a set of question–answer (QA) pairs. These QA pairs are typically yes/no format and represent expected outcomes if the video correctly exhibits the intended behavior, serving as explicit checks for the target dimension, as the example in Fig. 2.

We then feed both the video and the generated QA pairs into a frozen MLLM, which internally assesses whether the video content aligns with the expectations posed by the questions. Rather than returning specific answers, the MLLM directly outputs a scalar score reflecting the degree of agreement between the video and the reference answers. Videos that more closely match the dimension-specific world knowledge encoded in the QA pairs receive higher scores. This enables scalable, fine-grained evaluation without requiring human annotation. For a small set of dimensions that require fine-grained visual perception, we employ specialized pre-trained vision models like CoTracker2 (Karaev et al., 2024). Detailed descriptions of these modules can be found in the supplementary material.

**Optimization with Realism Constraints.** The resulting scores serve as sample-level rewards, which we use to dynamically adjust the training weight for each sample. Formally, the reward-weighted training loss is:

$$\mathcal{L}_{P_s}(\theta) = \mathbb{E}_{t,z_0,(z_1,x,p)\sim\mathcal{D}^{\mathrm{syn}}}\left[-r_\phi(x,p)\cdot||u(z_t,p,t;\theta) - v_t||^2\right], \tag{1}$$

where $z_1$ and $p$ are the clean latent of video $x$ and its caption. The formulation of standard flow-matching training, as is explained in Appendix A.1, is trained to predict velocity $u(z_t,p,t;\theta)$ from a random noise sample $z_0 \sim \mathcal{N}(0,I)$ to fit the ground-truth velocity $v_t$. We add the obtained score $r_\phi(x,p)$ given by the reward model as the loss weight. This training pipeline allows the model to place greater emphasis on high-quality or informative samples during training, while deprioritizing noisy or less useful ones. Notably, this effective training process is realized automatically, without requiring any human-labeled data.

To prevent the model from overfitting to synthetic distributions or drifting away from realistic video properties, we introduce a realism constraint. For prompts derived from captions of real videos (as discussed in Sec. 3.2), we generate corresponding synthetic videos using the same pipeline and include both the original and synthetic pairs during training. Additionally, we apply a KL divergence loss to encourage the model's outputs to remain consistent with the distribution of real videos.

$\hat{p}$ represents the real video's caption. $x$ and $z_1$ refer to a synthesized video generated from this caption and its latent embedding, respectively. With the predicted velocity $u(z_t,\hat{p},t;\theta)$, the predicted clean latent can be represented by $u(z_t,\hat{p},t;\theta) + z_0$, following the same formulation as Eq. 4. Formally, the training objective is:

Table 1: **Experiments on single dimension of VBench-2.0.** We choose Wan2.1-1.3B (Wan et al., 2025) as our baseline. We compare the evaluation results of our fine-tuned GigaVideo-1 with the baseline and 4 other recent state-of-the-art video generation models across 17 VBench-2.0 dimensions. A higher score indicates better performance in the corresponding dimension.

| Models | Human Anatomy | Human Clothes | Human Identity | Composition | Mechanics | Material | Thermotics | Multi-view Consistency | Dynamic Spatial Relationship |
|---|---|---|---|---|---|---|---|---|---|
| HunyuanVideo (Kong et al., 2024) | 88.58 | 82.97 | 75.67 | 43.96 | 76.09 | 64.37 | 56.52 | 43.80 | 21.26 |
| CogVideoX-1.5 (Yang et al., 2024c) | 59.72 | 87.18 | 69.51 | 44.70 | **80.80** | **83.19** | 67.13 | 21.79 | 19.32 |
| Sora (OpenAI, 2024) | 86.45 | **98.15** | **78.57** | **53.65** | 62.22 | 64.94 | 43.36 | 58.22 | 19.81 |
| Kling 1.6 (Kuaishou, 2024.06) | 86.99 | 91.75 | 71.95 | 43.89 | 65.55 | 68.00 | 59.46 | **64.38** | 20.77 |
| Wan2.1 (Wan et al., 2025) | 85.87 | 89.00 | 67.02 | 44.23 | 74.42 | 69.64 | 72.66 | 44.60 | 22.22 |
| GigaVideo-1 | **90.18**(↑4.31) | **95.10**(↑6.1) | 69.14(↑2.12) | 47.96(↑3.73) | 76.30(↑1.88) | 72.32(↑2.68) | **73.19**(↑0.53) | 52.87(↑8.27) | **26.57**(↑4.35) |

| Models | Dynamic Attribute | Motion Order Understanding | Human Interaction | Complex Landscape | Complex Plot | Camera Motion | Motion Rationality | Instance Preservation | Mean |
|---|---|---|---|---|---|---|---|---|---|
| HunyuanVideo (Kong et al., 2024) | 22.71 | 26.60 | 67.67 | 19.56 | 10.11 | 33.95 | 34.48 | 73.79 | 49.53 |
| CogVideoX-1.5 (Yang et al., 2024c) | 24.18 | 26.94 | 73.00 | **23.11** | **12.42** | 33.33 | 33.91 | 71.03 | 48.90 |
| Sora (OpenAI, 2024) | 8.06 | 14.81 | 59.00 | 14.67 | 11.67 | 27.16 | 34.48 | 74.60 | 47.64 |
| Kling 1.6 (Kuaishou, 2024.06) | 19.41 | 29.29 | 72.67 | 18.44 | 11.83 | **61.73** | 38.51 | **76.10** | 52.98 |
| Wan2.1 (Wan et al., 2025) | 46.15 | 29.97 | 72.33 | 17.11 | 10.69 | 36.11 | 40.80 | 63.26 | 52.12 |
| GigaVideo-1 | **49.08**(↑2.93) | **33.67**(↑3.70) | **80.67**(↑8.34) | 18.44(↑1.33) | 11.33(↑0.64) | 36.11(↑0) | **52.87**(↑12.07) | 69.03(↑5.77) | **56.17**(↑4.05) |

$$\mathcal{L}_{P_r}(\theta) = \mathbb{E}_{t, z_0, (\hat{z}_1, \hat{p}) \sim \mathcal{D}^{\text{real}}, (x, z_t) \sim \mathcal{D}^{\text{syn}}} \left[ -r_\phi(x, \hat{p}) \cdot ||u(z_t, \hat{p}, t; \theta) - v_t||^2 \right. \\ \left. + \lambda_{\text{kl}} \cdot D_{\text{KL}} \left( (u(z_t, \hat{p}, t; \theta) + z_0) \parallel \hat{z}_1 \right) \right], \tag{2}$$

where $\hat{z}_1$ represents the real video's latent vector. Here, the first term encourages the model to learn from controlled, failure-mode-driven synthetic videos, while the second term preserves distributional alignment with real data by minimizing the latent distance between the latent vector of a real video and the model-predicted latent vector for the corresponding synthetic video. This dual-objective loss enables efficient fine-tuning to challenging dimensions without compromising its overall generation quality.

Overall, the total training loss consists of two parts, as shown in the following equation.

$$\mathcal{L}(\theta) = \lambda_{p_s} \cdot \mathcal{L}_{P_s}(\theta) + \lambda_{p_r} \cdot \mathcal{L}_{Pr}(\theta), \tag{3}$$

where $\lambda_{p_s}$ and $\lambda_{p_r}$ are empirically set to 0.5 and 0.5.

# 4 EXPERIMENT

## 4.1 EXPERIMENTAL SETUPS

**Models and Settings.** We specify key hyperparameters and architectural choices here for reproducibility. We use the Qwen2.5-7B-Instruct (Yang et al., 2024a) model as our LLM for prompt augmentation and question generation for VQA, while LLaVA-Video-7B (Zhang et al., 2024b) serves as the reward model to score synthesized videos. As the baseline T2V generation model, we choose Wan2.1-T2V-1.3B (Wan et al., 2025), applying full-parameter fine-tuning on its transformer to improve its expressiveness. We set the target resolution of video generation to 832×480 and generate 81 frames per video. During training, the batch size is set to 4 and the $\lambda_{kl}$ is set to 0.3 empirically. The learning rate is $1e-6$, with a single training epoch. This lightweight training schedule is designed to avoid overfitting and to preserve the general-purpose generation capabilities of the pretrained model.

**Datasets and Evaluations.** The training data consists of two main sources, as shown in Sec. 3.2. The real-world dataset is sampled from the Koala (Wang et al., 2024a) dataset, and the synthetic dataset is generated by different pre-trained T2V models (Kong et al., 2024; He et al., 2024b; Kuaishou, 2024.06; OpenAI, 2024; Wan et al., 2025). As for the evaluation, GigaVideo-1 offers a flexible optimization framework adaptable to various capability dimensions. To demonstrate its versatility, we instantiate the framework on VBench-2.0's 17 evaluation dimensions as concrete application instances. More details about the data distribution and the selected benchmark can be found in the Appendix A.2 and B.

## 4.2 Evaluation Results

**Qualitative Comparison.** To evaluate the effectiveness of GigaVideo-1, we choose 17 dimensions from VBench-2.0 as instances of our targeted dimensions. As detailed in Tab. 1, various aspects like human attribute, commonsense, and physical plausibility are considered. We compare with our baseline Wan2.1 (Wan et al., 2025) and the other 4 state-of-the-art video generation models (Kong et al., 2024; Yang et al., 2024c; OpenAI, 2024; Kuaishou, 2024.06). Settings of these models can be found in the Appendix A.3. After applying our GigaVideo-1 fine-tuning pipeline, we observe substantial improvements over the baseline across most dimensions. Specifically, with only 4 GPU-hours of training for single dimension, our method achieves an average gain of approximately 4% across the 17 dimensions. Notably, we observe an 8.27% improvement in the Multi-view Consistency, 8.34% improvement in the Human Interaction dimension and an increase of over 12% in Motion Rationality. The average performance gain over Wan2.1 (Wan et al., 2025) is approximately 4%. While the five advanced T2V generation models exhibit different strengths and weaknesses across various aspects, our GigaVideo-1 achieves the best overall performance after fine-tuning, outperforming the previous state-of-the-art on average.

Moreover, as shown in Tab. 1, the improvements brought by GigaVideo-1 vary across different evaluation dimensions. While our method delivers substantial gains in most cases, the improvements are relatively modest on certain dimensions, such as Camera Motion. We hypothesize that it stems from two interconnected factors. First, to maintain realistic video characteristics, our training relies on synthetic data generated from real-world captions. While these captions naturally emphasize frequently occurring attributes like Motion Rationality and Human Interaction, they inherently underrepresent less common aspects, such as specific camera movements or complex physical interactions. Second, the assessment of some technically complex dimensions depends on specialized reward models (e.g., CoTracker2 for camera motion tracking), which employ significantly stricter scoring criteria compared to the VLM-based assessment used for other dimensions. This combination of a naturally low occurrence in real-world data and rigorous evaluation standards results in fewer valid training instances for these specific dimensions, ultimately constraining the performance improvement. We anticipate that proactively expanding the prompt pool within our data engine to better target these underrepresented dimensions could alleviate this limitation in future work.

**Quantitative Analysis.** We provide qualitative visualizations of generated videos in Fig. 3. As shown, GigaVideo-1 demonstrates clear improvements over the baseline, particularly on dimensions where the original model underperforms. The generated sequences, like the growth of an insect or the melting of a snowman, exhibit more realistic physical dynamics and align better with commonsense expectations. Scenes involving human interactions also appear more visually plausible. We additionally conduct a user study to assess the quality preference of videos generated by Wan2.1 and GigaVideo-1 across the aforementioned evaluation dimensions. Details of the evaluation protocol and results are provided in the Appendix D and https://anonymousgigavideo.github.io/.

## 4.3 Ablation Studies

We conduct ablation studies to assess the core design choices in GigaVideo-1, focusing on the contributions of the prompt-driven data engine and the reward-guided optimization strategy. Additional ablations are provided in the supplementary materials, including the cross-model backbone ablations (Appendix C.2) and realism constraint ablations (Appendix C.4).

**Impact of Prompt-Driven Data Enhancement.** We investigate the effectiveness of the prompt-driven data engine, as summarized in Tab. 2. Our training data is structured into two distinct categories by prompts: (1) synthetic data from LLM-crafted prompts targeting weaknesses $P_s V_s$, and (2) hybrid data from real captions $P_r V_s + P_r V_r$, which combines synthetic videos with real videos for realism constraints.

Using $P_s V_s$ alone improves performance by 4%, highlighting targeted weakness remediation. The hybrid data ($P_r V_s + P_r V_r$) yields a 7% gain, underscoring the importance of realism constraints from real videos. Naively blending $P_s V_s$ with only $P_r V_s$ (without $P_r V_r$) causes degradation to 73.66, due to distributional conflicts between synthetic and real prompts. Incorporating $P_r V_r$ mitigates this issue, improving to 79.33 by providing grounding. Our full pipeline of blending $P_s V_s$ and $P_r V_s + P_r V_r$ achieves the highest performance by leveraging complementary strengths: The addition of real videos

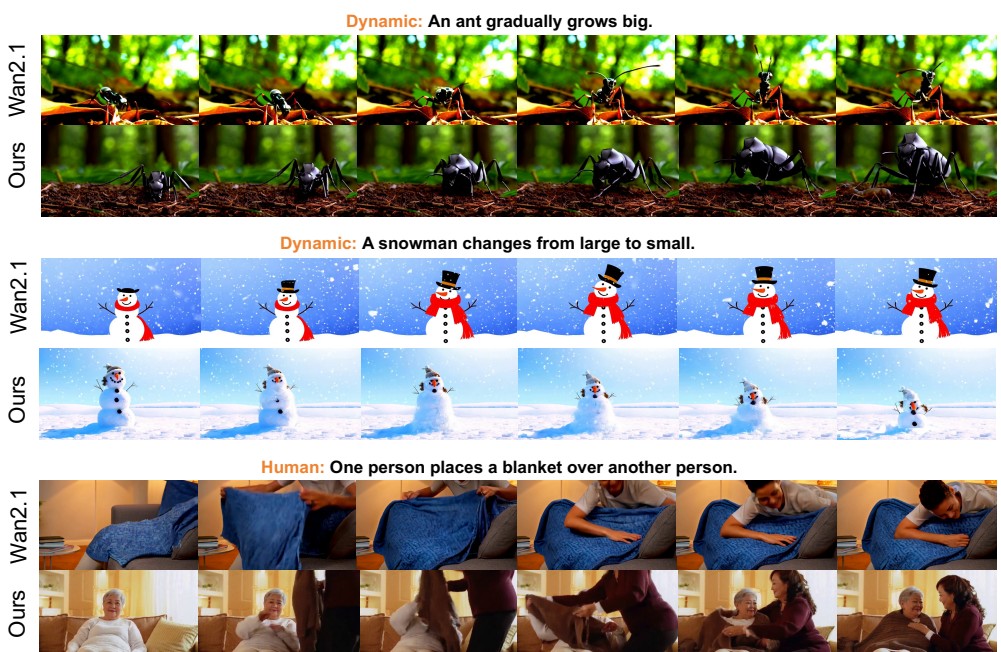

Figure 3: Visual comparisons of videos generated by GigaVideo-1 and Wan2.1 (Wan et al., 2025) across different real-world dimensions.

mitigates the potential bias introduced by synthetic prompts, while the synthetic components enhance coverage of model weaknesses in a controlled manner.

**Benefit of Reward-Guided Optimization.** Tab. 3 presents a systematic comparison of optimization strategies building upon the Wan baseline. We first evaluate supervised fine-tuning approaches using data generated from our prompt-driven engine. Full-data SFT with all generated samples improves accuracy to 75.67% but requires substantial computational resources (5.75 hours per epoch). To enhance efficiency, we filter out samples that cannot be reliably scored under the target dimension and those receiving zero reward from gradient updates. Applying SFT to this filtered dataset reduces training time to 0.90 hours per epoch while maintaining comparable accuracy (75.00%), confirming that data filtering alone does not yield significant performance gains. These SFT-based approaches train on the mixed-quality synthetic dataset uniformly. The negative influence of low-quality samples is averaged into the gradient updates, ultimately diluting the learning signal and potentially harming performance.

We then examine different reward integration strategies with distinct methodological characteristics in Tab. 3. The results reveal clear performance patterns. Online gradient-based methods achieve higher accuracy (79.33%/79.00%) but incur substantial computational overhead (∼7.5 hours/epoch), as backpropagating through the reward signal increases training burden despite marginal accuracy improvements. The RL-based Flow-GRPO approach reaches 79.67% accuracy but demands 650 hours per epoch, since it requires complete forward inference and probability calculations with its reference model. In contrast, our offline reweighting strategy achieves superior accuracy (80.67%) with minimal training time (0.90 hours/epoch). Implementation details and reason analysis can be found in Appendix C.1.

These comprehensive results validate GigaVideo-1's core innovations: our prompt-driven data engine generates targeted training samples, while offline reward integration enables stable, efficient optimization. This combination outperforms conventional SFT, gradient-based online methods, and RL-based approaches in both accuracy and computational efficiency, establishing a practical paradigm for scalable video model refinement.

Table 2: **Ablation of Data Engine**. "$P_r$" and "$P_s$" refer to using real video captions as prompts and generating metric-targeted prompts, respectively. "$V_r$" and "$V_s$" is using real and synthetic videos for reward labeling, respectively.

| $P_s V_s$ | $P_r V_s$ | $P_r V_r$ | Acc ↑ |
|---|---|---|---|
| | | | 72.33 |
| ✓ | | | 76.33 |
| | ✓ | | 79.00 |
| ✓ | ✓ | | 73.66 |
| | ✓ | ✓ | 79.33 |
| ✓ | ✓ | ✓ | 80.67 |

Table 3: **Ablation of Reward Strategy**. "SFT" means vanilla fine-tuning, while "filtered" means only using samples with a positive score. "Reweight / Backprop" means using the reward scores as loss weight or direct loss for backpropagation. "Online / Offline" subscripts refer to different scoring stages of reward models.

| Method | Type | Time(4GPU) ↓ | Acc ↑ |
|---|---|---|---|
| Wan | - | - | 72.33 |
| Wan+$SFT$ | Sft-based | 5.75h/epoch | 75.67 |
| Wan+$SFT(filtered)$ | | 0.90h/epoch | 75.00 |
| Wan+$Reweight_{online}$ | Grad-based | 7.74h/epoch | 79.33 |
| Wan+$Backprop_{online}$ | | 7.50h/epoch | 79.00 |
| Wan+$FlowGRPO_{online}$ | RL-based | 650h/epoch | 79.67 |
| Wan+$Reweight_{offline}$ | Offline | 0.90h/epoch | 80.67 |

Table 4: **Combined Enhancement of Different Dimensions.** We combine the training data of related dimensions. 4 different combinations are included.

| Aspect | Include Dimensions | | | | | Aspect | Include Dimensions | | | |
|---|---|---|---|---|---|---|---|---|---|---|
| ① Human | **HAn** | **HCl** | **HIn** | - | **Mean** | ② Physics | **Mec** | **The** | **Mat** | **Mean** |
| Wan2.1 | 85.87 | 89.00 | 72.33 | - | 82.40 | Wan2.1 | 74.42 | 72.66 | 69.64 | 72.24 |
| GigaVideo-1 | **90.18** | **95.10** | **80.67** | - | **88.65** | GigaVideo-1 | **76.30** | **73.19** | 72.32 | **73.94** |
| Combined | 88.21 | 89.90 | 76.67 | - | 84.93 | Combined | 75.18 | 69.57 | **75.89** | 73.55 |
| ③ Dynamics | **DSR** | **MOU** | **MoR** | **DAt** | **Mean** | ④ Exisitence | **Com** | **IPr** | - | **Mean** |
| Wan2.1 | 22.22 | 29.97 | 40.80 | 46.15 | 34.79 | Wan2.1 | 44.23 | 63.26 | - | 53.75 |
| GigaVideo-1 | 26.57 | 33.67 | **52.87** | 49.08 | 40.55 | GigaVideo-1 | 47.96 | 69.03 | - | 58.50 |
| Combined | **26.87** | **37.50** | 47.70 | **55.68** | **41.94** | Combined | **50.74** | **70.80** | - | **60.77** |

## 4.4 COMBINED ENHANCEMENT OF DIFFERENT DIMENSIONS

Beyond evaluating the effectiveness of our pipeline on individual dimensions, we further explore interactions and joint optimization across multiple dimensions, as shown in Tab. 4. We begin by grouping semantically related dimensions into 4 broader categories based on conceptual overlap: Human, Dynamic, Physics, and Existence. For each group, we use multi-dimensional reward scores to jointly train the model across all dimensions within that group.

As shown in Tab. 4, the results vary across groups. For the Human and Physics categories, joint training does not outperform single-dimension tuning on most metrics. In contrast, for Dynamic and Existence, joint optimization leads to consistent gains across the majority of constituent dimensions. We hypothesize that the limited gains in the Human and Physics categories arise from conflicting objectives across their dimensions. While semantically related, these dimensions often target diverse aspects that may interfere during joint training. In contrast, dimensions within the Dynamic and Existence groups tend to have more consistent goals, allowing for more effective joint optimization. These findings highlight the need for clearer dimension definitions and more deliberate prompt design to ensure aligned supervision across multiple evaluation axes.

## 5 CONCLUSION

We present *GigaVideo-1*, a lightweight fine-tuning framework that significantly enhances text-to-video generation. By leveraging our prompt-driven data engine, our method effectively improves video quality by targeting underperforming aspects such as physical consistency and temporal logic. This approach ensures more focused and efficient model adaptation, without the need for large-scale data collection or manual annotations. Secondly, the proposed reward-guided optimization utilizes automatic feedback to further refine model performance. Adaptively adjusting the training process leads to consistent and realistic outputs. These two innovations enable GigaVideo-1 to achieve

substantial improvements in video generation quality while minimizing computational cost and data requirements, offering a scalable and practical solution for advancing video generation models.

## REPRODUCIBILITY STATEMENT

To ensure reproducibility, we detail our model's architecture, datasets, training, and hyperparameters in Sec. 4.1 and Appendix A. Moreover, the pre-trained models and data we utilized are all open-source and easily accessible.

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

## A IMPLEMENTATION DETAILS

### A.1 PRELIMINARY OF T2V TRAINING FORMULATION

This appendix details the standard training formulation for text-to-video models, which serves as the foundation for the GigaVideo-1 pipeline described in Sec. 3.

Recent advances in video generation (Esser et al., 2023; Brooks et al., 2024; Chen et al., 2023; Wang et al., 2023a) have shifted toward principled training formulations. Among them, flow-matching has emerged as an effective approach for training diffusion-based models. It frames generation as learning a continuous transformation from noise to data through ordinary differential equations (ODEs), offering both stability and theoretical alignment with maximum likelihood training.

In this setting, training involves linearly interpolating between a random noise sample $z_0 \sim \mathcal{N}(0, I)$ and a clean video latent $z_1$. Following Rectified Flows (RFs) (Esser et al., 2024), the model is trained to predict the ground-truth velocity $v_t$ between these two endpoints. The intermediate latent $z_t$ and the ground-truth velocity $v_t$ are formed as:

$$z_t = tz_1 + (1-t)z_0, \quad v_t = \frac{dz_t}{dt} = z_1 - z_0. \tag{4}$$

The learning objective minimizes the discrepancy between the predicted and actual velocity $v_t$, conditioned on the prompt embedding and timestep:

$$\mathcal{L} = \mathbb{E}_{t, z_0, (z_1, p) \sim \mathcal{D}^{\mathrm{real}}} ||u(z_t, p, t; \theta) - v_t||^2, \tag{5}$$

where $p$ is the text-form prompt, $\theta$ is the model weights, and $u(z_t, p, t; \theta)$ denotes the output velocity predicted by the model.

This formulation reveals two core factors that directly impact fine-tuning performance: 1) The composition of training samples determines what types of visual behaviors the model will learn to refine, making it crucial to expose the model to failures it needs to correct. 2) The optimization strategy dictates how each sample contributes to training. A static formulation treats all samples equally, while prioritizing more informative or underperforming cases can be more effective. These two perspectives form the basis of the fine-tuning pipeline introduced in the main text.

### A.2 EVALUATION DETAILS.

As a recently proposed benchmark that systematically investigates the limitations of current video generation models. VBench-2.0 decomposes overall video generation quality into 18 fine-grained, hierarchical dimensions, providing tailored prompts and evaluation protocols for each. Since the "diversity" dimension cannot be meaningfully assessed on a single synthesized video, we exclude it from our pipeline. The remaining 17 dimensions are used as our fine-tuning targets and serve as the benchmark for evaluating model performance.

### A.3 MODEL SETTINGS.

We compare the evaluation results of our fine-tuned GigaVideo-1 with the baseline and 4 other recent state-of-the-art video generation models across 17 VBench-2.0 dimensions. A higher score indicates better performance in the corresponding dimension. Details of the 6 state-of-the-art T2V models in Tab. 4 can be found in Tab. 5.

Table 5: **Information on Evaluated Models.**

| Model Name | Video Length | Per-Frame Resolution | Frame Rate (FPS) | Frame Count |
|---|---|---|---|---|
| HunyuanVideo (Kong et al., 2024) | 5.3s | 720×1280 | 24 | 161 |
| CogVideoX-1.5 (Yang et al., 2024c) | 10.1s | 768×1360 | 16 | 129 |
| Sora-480p (OpenAI, 2024) | 5.0s | 480×854 | 30 | 150 |
| Kling 1.6 (Kuaishou, 2024.06) | 10.0s | 720×1280 | 24 | 241 |
| Wan2.1 (Wan et al., 2025) | 5.0s | 480×832 | 16 | 81 |
| GigaVideo-1 | 5.0s | 480×832 | 16 | 81 |

## A.4 REWARD MODELS

Given that MLLMs are already capable of accurately understanding video scenes and events in most cases, we use them to score the performance of synthesized videos across the majority of evaluation dimensions. However, current MLLMs exhibit limitations in fine-grained visual perception, making their scores less reliable for a few specific dimensions that require such precision. For these cases, we employ specialized pre-trained visual models to provide more accurate assessments. For instance, in the instance preservation dimension, we use YOLO-World for object detection and counting; in the camera motion dimension, we use Cotracker2 for point tracking. As more capable MLLMs or scoring models become available in the future, the scoring module of GigaVideo-1 can be directly upgraded, further enhancing the reliability of reward-guided optimization.

## B TRAINING DATA.

### B.1 DATASET OVERVIEW.

The training data comprises two primary sources, as described in Sec. 3.3. First, the real-world dataset includes approximately 3.5k high-quality video samples selected from the Koala (Wang et al., 2024a) dataset. Each sample is paired with an accurate and detailed caption, along with a corresponding synthetic video generated using that caption. To align with the duration of training data used in Wan2.1, we retain only Koala videos shorter than 5 seconds. Further filtering is conducted using the Video Training Suitability Score provided by Koala, which is computed via the Training Suitability Assessment Network proposed in its paper. Second, the synthetic dataset consists of approximately 9.5k videos generated from domain-specific prompts produced by our prompt-driven data engine. These prompts are designed to emphasize specific generative aspects and support targeted fine-tuning across diverse quality dimensions.

### B.2 PROMPT GENERATION

We start by identifying specific failure modes in the pre-trained model (e.g., unrealistic motion, implausible interactions) and selecting a few representative prompt examples that expose these issues. Each weak dimension is defined with a short description and a handful of seed phrases. These are then fed into a large language model using few-shot prompting to generate a diverse set of base prompts that are concise, visually grounded, and explicitly focused on the target concept (e.g., "Camera zoom in, Disneyland.").

> **Motion Order Understanding** evaluates whether models can generate multiple actions in a specified temporal order. To support this evaluation, we design base prompts for video generation that are concise, visually grounded, and explicitly focused on this concept. Each prompt should describe a clear transition between two distinct actions. For example:
> • A dog barks once, then suddenly runs to chase a toy.
> • A child waves at the camera, then quickly turns to run away.
> • A horse is grazing in the field, then it suddenly starts running across the meadow.
> These examples illustrate the intended structure. All prompts should depict two sequential actions that can reasonably occur within a 3–5 second video. Repetition of the above examples should be avoided, and each prompt must involve an explicit and coherent motion transition. Output 100 prompts directly.

Figure 4: LLM prompts we use in the first stage for analogy and sentence making, targeting the Motion Order Understanding dimension as an example.

Specifically, we provide an example prompt targeting the Motion Order Understanding dimension to illustrate the generation process of the base prompts, as shown in the Fig. 4. For the second augmentation stage, in order to maintain consistency with the data distribution of the pre-trained model, we adopt the same prompt augmentation template provided by Wan2.1.

### B.3 DATA DISTRIBUTION

Tab. 6 summarizes the prompt distributions and data allocation across the 17 VBench-2.0 dimensions. GigaVideo-1 primarily relies on synthetically generated data for each dimension to secure broad prompt coverage and stable controllability, with real-world data serving as a minimal constraint.

Table 6: **Data Distributions.** "SynP Rate" denotes the ratio of synthetic to real-video prompts within each dimension's training data. "Dim Rate" denotes the proportion of each dimension's data relative to the total across all 17 dimensions.

| Models | Human Anatomy | Human Clothes | Human Identity | Composition | Mechanics | Material | Thermotics | Multi-view Consistency | Dynamic Spatial Relationship |
|---|---|---|---|---|---|---|---|---|---|
| SynP Rate | 83.0% | 81.2% | 85.8% | 89.7% | 81.4% | 82.5% | 80.6% | 88.3% | 90.0% |
| Dim Rate | 23.7% | 16.6% | 8.5% | 2.3% | 2.1% | 1.7% | 1.3% | 3.6% | 1.0% |

| Models | Dynamic Attribute | Motion Order Understanding | Human Interaction | Complex Landscape | Complex Plot | Camera Motion | Motion Rationality | Instance Preservation | - |
|---|---|---|---|---|---|---|---|---|---|
| SynP Rate | 91.8% | 94.0% | 81.4% | 92.7% | 90.8% | 93.1% | 80.5% | 83.0% | - |
| Dim Rate | 1.5% | 1.5% | 3.4% | 1.2% | 2.2% | 0.9% | 13.9% | 14.5% | - |

While we considered data balancing across dimensions, significant variation in data construction difficulty led to an inherent imbalance. During quality assessment, videos generated from targeted prompts were frequently rejected when failure modes became indistinguishable, resulting in differing valid-data proportions. A natural long-tail pattern is exhibited in which human-centric axes occupy more mass, whereas geometry, physics, and structural dimensions are lighter. This final distribution follows practical considerations such as data discoverability, licensing of real-video sources, and the greater difficulty of specifying and verifying rare or complex phenomena, and it aligns with quality filtering that tends to favor reliably instantiated prompts.

## C  MORE ABLATIONS

### C.1  REWARD ABLATION IMPLEMENTATION OF TAB. 3

We examine different reward integration strategies with distinct methodological characteristics in Tab. 3. The "Reweight-offline" approach uses offline scores computed from synthetic videos prior to training. "Reweight-online" predicts reward scores during training based on intermediate denoised frames, while "Backprop-online" directly incorporates reward scores into the loss function, enabling gradient backpropagation through the reward signal. For fair comparison with RL-based methods, we implement Flow-GRPO using the same MLLM as the reward function and targeted prompts from our data engine. In online and RL methods, we simplify the reward participation process by employing the MLLM directly to score generated videos, omitting the intermediate Q&A generation step for efficiency considerations.

The performance advantage of GigaVideo-1 stems from fundamental methodological differences. Online methods suffer from scoring inconsistencies due to the variable visual fidelity of intermediate denoised frames used during training. Furthermore, direct MLLM scoring without structured Q&A pairs often fails to maintain consistent quality assessments across samples. Our offline approach circumvents these issues through precomputed reliable scores and focused dimension-specific optimization, demonstrating an optimal balance between efficiency and effectiveness for practical video generation enhancement.

### C.2  CROSS-MODEL BACKBONES

We further validate the GigaVideo-1 framework by applying it to diverse text-to-video backbones with varying model sizes and configurations. As shown in Tab. 7, the framework brings consistent performance improvements across all baseline models, though the magnitude of gain varies due to differences in model capacity and architectural settings. This result demonstrates the robustness of our approach across heterogeneous base architectures, confirming that its effectiveness does not rely on specific model idiosyncrasies.

The framework achieves this architecturally invariant efficiency through several key design features: targeted training data generated for specific weaknesses makes model training more focused; offline scoring with pre-computed VLM feedback eliminates runtime overhead; and goal-directed optimization guided by scores concentrates updates on the most critical dimensions. These structural properties

Table 7: **Ablation of cross-model backbones**. All results are evaluated on the Human Interaction dimension of VBench-2.0.

| Model | Size | Length | Resolution | FPS | Frames | Baseline | GigaVideo-1 | △ | Time(4GPU) |
|-------|------|--------|------------|-----|--------|----------|-------------|---|------------|
| LTXVideo | 2B | 6.7s | 512×768 | 24 | 161 | 34.33 | 51.67 | +17.34 | 0.61h/epoch |
| CogVideoX | 5B | 6s | 480×720 | 8 | 49 | 60.67 | 70.33 | +9.66 | 2.23h/epoch |
| HunyuanVideo | 13B | 5.3s | 720×1280 | 24 | 161 | 67.67 | 76.00 | +8.33 | 6.71h/epoch |
| Wan2.1 | 1.3B | 5.0s | 480×832 | 16 | 81 | 72.33 | 80.67 | +8.34 | 0.90h/epoch |

collectively ensure stable computational advantages while delivering quality gains proportional to each model's baseline capabilities.

## C.3 DIMENSION-WISE GENERALIZABILITY ON A DIFFERENT BENCHMARK

While VBench 2.0 served as the primary benchmark for its rigor and acceptance, GigaVideo-1 is not inherently tied to VBench 2.0's specific dimensions. The core design accepts any dimension definition as input, enabling adaptability to diverse benchmarks.

Table 8: **Experiments on single dimension of VMBench**. We choose Wan2.1-1.3B 352x640, 93 frames as baseline. All results are evaluated on the 5 dimensions of VMBench. CAS, MSS, OIS, PAS, and TCS refer to Common Sense, Motion Smoothness, Object Integrity, Perceivable Amplitude, and Temporal Consistency, respectively.

| Model | CAS | MSS | OIS | PAS | TCS | Avg |
|-------|-----|-----|-----|-----|-----|-----|
| **Wan2.1** | 62.8 | 84.2 | 66.0 | 17.9 | 97.8 | 65.7 |
| **GigaVideo-1** | 64.5 | 84.0 | 69.3 | 28.9 | 98.3 | 69.0 |

GigaVideo-1's design can seamlessly integrate any new evaluation dimension that can be textually defined and assessed by an MLLM. To preliminarily demonstrate this, we conducted experiments on VMBench, another benchmark with different dimension definitions. The results in 10 show positive effects, though the improvement of some dimensions is less pronounced due to holistic and interwoven dimension definitions, proving our broad applicability.

## C.4 REALISM CONSTRAINTS.

Tab. 9 examines the effect of the realism constraint $\lambda_{KL}$ on fidelity, diversity, and task accuracy, evaluated on the Human Interaction dimension. We compute FVD between synthetic videos (Human Interaction dimension) and real Koala_HI videos to quantify the distributional discrepancy. The reference value refers to the self-similarity baseline of real Koala_HI videos. As $\lambda_{KL}$ increases from 0 to 0.3, FVD consistently improves and accuracy rises correspondingly, suggesting that a moderate realism penalty effectively aligns synthetic videos with real-data statistics without compromising utility. Beyond $\lambda_{KL}$=0.3, stronger constraints degrade both fidelity and accuracy, indicating over-regularization.

Diversity in the above table is computed by sampling 20 videos per text prompt and measuring inter-sample variation using VGG19-derived style/content metrics. The reference value shows the diversity of the generated videos of baseline Wan2.1. Diversity remains stable across all settings, demonstrating that moderate penalties do not induce mode collapse. The optimal setting ($\lambda_{KL}$=0.3) most effectively narrows the realism gap while preserving diversity, establishing a practical operating point for realism-constrained training.

Table 9: **Ablation of realism constraints**. FVD is computed between synthetic videos and real Koala_HI videos, and "Ref" reports the self-similarity of the real videos. All results are evaluated on the Human Interaction dimension of VBench-2.0.

| $\lambda_{KL}$ | 0 | 0.05 | 0.1 | 0.3 | 0.5 | 1.0 | Ref |
|---|---|---|---|---|---|---|---|
| FVD ↓ | 1752.5 | 1601.9 | 1557.9 | 1541.8 | 1683.4 | 1943.1 | 1164.6 |
| Diversity ↑ | 52.9 | 52.7 | 52.8 | 52.7 | 52.3 | 52.1 | 52.8 |
| Acc ↑ | 73.66 | 75.00 | 79.00 | 80.67 | 77.00 | 73.67 | 72.33 |

## C.5 IMPACT ON NON-OPTIMIZED VIDEO QUALITY

To evaluate the impact of one dimension on the others, we fine-tune a model on Human Interaction (HI) and assess its performance on the remaining dimensions.

Table 10: **Single-dimension experiments on VBench.** We use the HI–specific fine-tuned model to assess performance on other dimensions. All results are reported on the five VBench dimensions.

| Model | Background Consistency | Aesthetic Quality | Subject Consistency | Motion Smoothness | Dynamic Degree |
|---|---|---|---|---|---|
| Wan2.1 | 94.09 | 56.97 | 93.55 | 98.58 | 79.67 |
| GigaVideo-1 | 93.58 | 52.72 | 92.21 | 98.60 | 93.67 |

As shown in Tab 10, fine-tuning on the Human Interaction (HI) dimension yields substantial gains on Dynamic Degree while largely preserving performance on non-targeted dimensions. In particular, GigaVideo-1 improves Dynamic Degree from 79.67 to 93.67 (+14.0), indicating that the HI-oriented training significantly enhances the richness of motion in the generated videos. Meanwhile, Background Consistency (94.09 → 93.58), Subject Consistency (93.55 → 92.21), and Motion Smoothness (98.58 → 98.60) remain effectively unchanged, showing no noticeable degradation. We observe a moderate trade-off on Aesthetic Quality (56.97 → 52.72), suggesting that targeted optimization primarily affects motion expressiveness while keeping structural and temporal coherence intact.

# D USER STUDY

We additionally conduct a user study to assess the quality preference of videos generated by Wan2.1 and GigaVideo-1 across the aforementioned evaluation dimensions. For each evaluation dimension, we randomly select five prompts and generate paired video samples using both Wan2.1 and GigaVideo-1 based on the same prompt. This results in pairwise comparison options, where the source model of each video is hidden and the order of presentation is randomized to prevent bias.

Table 11: **User Study of Different Dimensions.** This table presents evaluation results for our baseline Wan2.1 and GigaVideo-1 across 17 VBench-2.0 dimensions. A higher score indicates higher user preference in the corresponding dimension.

| Models | Human Anatomy | Human Clothes | Human Identity | Composition | Mechanics | Material | Thermotics | Multi-view Consistency | Dynamic Spatial Relationship |
|---|---|---|---|---|---|---|---|---|---|
| GigaVideo-1>Wan2.1 | 79.0% | 82.5% | 90.0% | 92.5% | 87.5% | 85.0% | 87.5% | 92.5% | 97.5% |

| Models | Dynamic Attribute | Motion Order Understanding | Human Interaction | Complex Landscape | Complex Plot | Camera Motion | Motion Rationality | Instance Preservation | Mean |
|---|---|---|---|---|---|---|---|---|---|
| GigaVideo-1>Wan2.1 | 85.0% | 90.0% | 82.5% | 95.0% | 100.0% | 92.5% | 80.0% | 95.0% | 88.8% |

Annotators are shown the name of the target dimension and asked to choose which video they prefer with respect to that specific aspect. Each dimension is evaluated by 10 annotators, and we compute the user preference score of GigaVideo-1 over Wan2.1 as the percentage of comparisons where GigaVideo-1 is preferred. As shown in Tab. 11, GigaVideo-1 achieves preference scores above 50%

on the majority of dimensions, demonstrating clear improvements in perceptual quality brought by our reward-guided optimization.

# E ALL DIMENSIONS EVALUATION

Beyond evaluating the effectiveness of our pipeline on individual dimensions and some related dimensions, we further explore the joint optimization across all 17 dimensions defined by VBench-2.0, as shown in Tab 12.

Table 12: **Combined Enhancement of All 17 Dimensions of VBench-2.0.** This table presents evaluation results of our baseline Wan2.1 and the combined enhancement version of GigaVideo-1 across 17 VBench-2.0 dimensions. A higher score indicates better performance of this dimension.

| Models | HAn | HCl | HId | HIn | DSR | MOU | MoR | DAt | Mec | The | Mat | Com | IPr | CMt | MVC | CoL | CoP | Mean |
|---|---|---|---|---|---|---|---|---|---|---|---|---|---|---|---|---|---|---|
| Wan 2.1 (Wan et al., 2025) | 85.87 | 89.00 | 67.02 | 72.33 | 22.22 | 29.97 | 40.80 | 46.15 | **74.42** | **72.66** | 69.64 | 44.23 | 63.26 | **36.11** | **44.60** | 17.11 | **10.69** | 52.12 |
| All | **89.88** | **91.26** | **67.33** | **73.33** | **29.47** | **37.71** | **51.15** | **47.99** | 70.53 | 65.94 | **75.22** | **49.78** | **71.41** | 35.49 | 33.55 | **22.22** | 10.36 | **54.27** |

Joint optimization generally yields good performance across most dimensions compared to our baseline Wan2.1. However, certain dimensions exhibit diminished gains. We hypothesize that this may be due to conflicting objective signals arising from inconsistent definitions or incompatible prompt formulations across different dimensions. Addressing this issue may require more principled dimension definitions and prompt engineering strategies to ensure coherent supervision across multiple axes of evaluation.

# F ADVANTAGE AND VALIDATION OF MLLM-BASED REWARD

Our framework's adoption of MLLM-based reward models addresses fundamental limitations in scalability, consistency, and physical reasoning capabilities compared to traditional human-supervised approaches. This section presents the strategic advantages of our approach and provides empirical validation addressing key concerns about its reliability.

## F.1 OPERATIONAL AND SCALABILITY ADVANTAGES

The MLLM-based reward generation completely eliminates the need for human-annotated training data, enabling rapid, large-scale assessment of generated content. This efficiency breakthrough makes targeted dimensional optimization practically feasible where it was previously economically prohibitive. Beyond mere efficiency gains, MLLM-based rewards demonstrate superior capability in handling complex physical and logical reasoning tasks by leveraging their extensive training on scientific knowledge. Furthermore, our approach features built-in evolutionary capacity: as foundation models advance, our reward quality automatically improves without requiring architectural changes.

## F.2 EMPIRICAL VALIDATION THROUGH HUMAN STUDY

To address concerns regarding the correlation between MLLM scores and human judgment, we conducted a comprehensive user study comparing MLLM assessments with human evaluations across 17 dimensions. The study design involved selecting three prompts per dimension, generating pairwise video comparisons, and having 15 annotators evaluate each pair. The results demonstrate strong human-MLLM alignment, with agreement rates exceeding 85% on most dimensions. Specific dimensions such as Motion Rationality (97.8%) and Dynamic Spatial Relationship (95.6%) showed particularly high consistency, while more complex dimensions like Mechanics (77.8%) reflected the inherent challenges in evaluating physical plausibility.

## F.3 DESIGN CONSIDERATIONS FOR MLLM BIAS

We acknowledge that MLLMs trained on web data are not perfect arbiters of physical truth. However, our framework transforms this limitation into a strength through deliberate design choices. First,

Table 13: Human-MLLM Agreement Rates Across Dimensions(%)

| Dimension | HI | Mat | MOU | TH | Mec | DSP | DA | HC | HA |
|-----------|------|------|------|------|------|------|------|------|------|
| Agreement | 82.2 | 86.7 | 97.8 | 75.6 | 77.8 | 95.6 | 93.3 | 91.1 | 93.3 |
| Dimension | MR | HI | CM | CL | CP | IP | MVC | Com | Avg |
| Agreement | 88.9 | 100 | 95.6 | 88.9 | 91.1 | 88.9 | 84.4 | 86.7 | 89.3 |

MLLMs provide significantly more structured understanding of physical commonsense than the base video generation models, creating a stable supervision signal Xue et al. (2025); Yin et al. (2025); Wu et al. (2025); Waseem & Shahzad (2024). Second, we incorporate explicit distributional anchoring through KL-divergence constraints that tether the optimization process to real video distributions, preventing synthetic bubble effects. This integrated approach ensures our method remains grounded in real-world physical interactions while leveraging the scalability of automated assessment.

## G    BROADER IMPACT

GigaVideo-1 presents a scalable and efficient pipeline for enhancing text-to-video generation, offering improved alignment with human preferences through automated reward modeling and dimension-specific fine-tuning. By significantly reducing the reliance on manual annotation and leveraging multimodal feedback, GigaVideo-1 lowers the barrier for high-quality video generation and has the potential to benefit a wide range of applications, including digital storytelling, educational content creation, and simulation prototyping. Several potential risks warrant attention. First, the reliance on pretrained foundation models means that GigaVideo-1 may inherit biases present in its underlying components, including in the scoring signals used for reward optimization. This could lead to unintended reinforcement of stereotypes or systematic errors if not properly audited. Second, this increased accessibility may bring the risk of misuse. Like other generative video technologies, GigaVideo-1 could be exploited to produce misleading or manipulated content, such as fabricated news footage or synthetic evidence. To mitigate this, it is important to promote responsible deployment practices, encourage transparency in generated media, and support ongoing research into detection and watermarking tools. Lastly, while our method improves specific dimensions of generation quality, it may lead to trade-offs in aspects not explicitly optimized, highlighting the importance of balanced, multi-objective evaluation and ongoing community benchmarking efforts. We encourage future research to explore more robust alignment signals, diversified data sources, and transparent evaluation protocols to ensure that methods like GigaVideo-1 contribute to the responsible and inclusive advancement of video generation technologies.

## H    LIMITATION AND FUTURE WORK

### H.1    DATA-DRIVEN WEAK DIMENSION DISCOVERY

A primary and promising direction is the development of a closed-loop, data-driven system for automatic weak dimension discovery. Future research will focus on replacing manual analysis with systematic methods. For example, an automated system may use a large set of diverse prompts and generate videos. Using MLLM as a failure detector and LLM-driven clustering is a possible pipeline for automated weakness discovery. We did a simple exploration to validate the viability of a fully automated loop. When explicitly instructed that "the video may contain issues and please identify them," although with some degree of hallucination, the MLLM can relatively accurately pinpoint failures (e.g., motion artifacts or implausible objects) in low-scoring videos. This demonstrates the feasibility of automated extraction.

This evolution would transform our framework into a self-improving system, capable of dynamically identifying and addressing its own weaknesses. Such an automated refinement loop is inherently complementary to our proposed pipeline and could be seamlessly integrated into systems like GigaVideo-1, substantially boosting overall robustness and adaptability without structural changes. We envision this as the key step towards achieving truly scalable and autonomous video generation models.

## H.2 DYNAMIC LOSS BALANCING

While our framework demonstrates effective fine-tuning capabilities, the current implementation employs static weighting for multi-dimensional joint training, which may lead to interference between competing objectives. This represents a meaningful limitation, particularly for dimensions with conflicting optimization paths. Looking forward, the integration of dynamic loss-balancing techniques, such as Grad-Norm, uncertainty weighting, or Pareto optimization, presents a promising avenue for mitigating such interference. Implementing these advanced multi-task learning strategies would allow the framework to automatically adjust loss weights during training, potentially resolving the conflicts observed between certain dimension groups. We thank the reviewer for this valuable suggestion and will pursue this direction to enhance the robustness and scalability of multi-dimensional video model optimization.

## I THE USE OF LARGE LANGUAGE MODELS

Large Language Models (LLMs) are only used for polishing and improving the clarity of the manuscript. All research-related work is conducted independently by the authors.

