# OpenReview forum: "GigaVideo-1: Advancing Video Generation via Automatic Feedback with 4 GPU-Hours Fine-Tuning"
_ICLR.cc/2026/Conference — Submitted to ICLR 2026_

### Official Review · Reviewer_YYeF · 2025-10-14

**Soundness:** 3
**Presentation:** 2
**Contribution:** 2
**Rating:** 4
**Confidence:** 3

**Summary:**

This paper presents an algorithm for the rapid scaling of video diffusion models, reporting up to a 4% performance gain with only 4 GPU hours. The approach involves two main components: a prompt-driven data engine to generate diverse synthetic training data and a VLM-based scoring mechanism to weight this data during training.

However, I am puzzled by the formulation of the proposed KL divergence constraint. The authors frame the objective as minimizing the distance between $u(z_t,p,t;\theta)+z_0$ and $z_1$. This is mathematically equivalent to minimizing the distance between the model's direct output, $u(z_t,p,t;\theta)$ and the target $z_1-z_0$. Given this equivalence, the decision to add $z_0$ to the model's prediction before computing the loss appears superfluous.

**Strengths:**

1. High Novelty: The core idea of actively identifying, synthesizing, and then reweighting challenging videos is highly innovative. This "target-seeking" optimization strategy, which explicitly pushes the model beyond its current capabilities, is a new direction for training diffusion models. To the best of my knowledge, no prior work has taken a similar approach.

2. Thorough Experimental Validation: The experiments are comprehensive and well-designed. The ablation studies are particularly insightful, systematically justifying the components of the proposed method. The authors have diligently tested their reweighting strategy across various training paradigms (including SFT, RL, and gradient-based methods), concluding that the offline reweighting approach is most effective.

3. Strong Empirical Results: The method achieves impressive results. Notably, the fact that this approach outperforms a strong baseline like Flow-GRPO provides compelling evidence for its effectiveness and practical value.

**Weaknesses:**

1. Request for Intuition on Method's Effectiveness: I find the core mechanism of the paper's success to be somewhat perplexing, despite the strong empirical results. The methodology involves training on videos synthesized from challenging prompts. Intuitively, videos generated for highly challenging prompts would be of lower quality, receive a lower VLM score, and therefore contribute less to the training objective due to the reweighting. It is therefore surprising that this strategy yields such a significant (nearly 5-point) improvement over standard SFT. Could the authors provide a more detailed explanation or intuition for why this approach is so effective?

2. Clarification on Data Blending in Table 2: Regarding the experiments in Table 2, when the datasets PsVs, PrVs, and PrVr are used concurrently, does this mean they are simply blended together for training? If so, what is the anticipated effect of also incorporating the PsVr dataset into this mixture?

3. Suggestion for Showcasing Qualitative Results: The static figures presented in the paper are not sufficient to fully demonstrate the superiority of the proposed method's video generation quality. While the supplementary videos are helpful, I would strongly suggest that the authors create an anonymous GitHub Pages site or a similar web-based platform. This would allow reviewers to more easily and directly compare the results and appreciate the qualitative improvements

**Questions:**

No

---

> ### Author Response · Authors · 2025-11-21
> **Response to Reviewer YYeF (1)**
>
> Thank you for your positive feedback on the high novelty, thorough experiments, and strong results of the proposed framework. In response to your question, we provide a detailed point-by-point response below.
>
> &nbsp;
>
> ---
>
> ### **W1:**
>
> Thank you for this profound question. Your observation on the mathematical equivalence is correct and helped us identify a notational error, for which we apologize.
>
>
>
> In our formulation, the KL divergence target was mis-specified. The objective should minimize the distance between the **model-predicted latent state** $(u(\cdot) + z_0)$ for the synthetic video and the **real video's latent state** $\hat{z}_1$, not $z_1$. Thus, the corrected term is:
>
> $D_{\text{KL}}\left( (u(z_t, \hat{p}, t; \theta) + z_0) \parallel \hat{z}_1 \right)$
>
> Correspondingly, the variable sources in Equation (2) should be corrected: the hat notation $\hat{z}_1$ and $\hat{p}$ denote values from the real dataset $\mathcal{D}^{\mathrm{real}}$, while the non-hat notation $(x, z_t)$ denotes values from the synthetic dataset $\mathcal{D}^{\mathrm{syn}}$.
>
>
>
> This design intentionally adds $z_0$ to preserve the **physical meaning of the realism constraint**. It ensures we constrain the model's **complete predicted latent representation** directly against real content, rather than just its update direction. While mathematically equivalent to minimizing $u(\cdot)$ against ($\hat{z}_1−z_0$), our formulation maintains conceptual clarity by explicitly aligning the full output with the real data distribution.
>
> The approach cleanly separates objectives: the reward term targets weakness-specific improvement, while the KL term anchors the model to realism. Adding $z_0$ is a deliberate choice to enforce this anchor clearly and effectively.
>
> We have corrected this in the manuscript on ***page 6, line 293***, and appreciate your feedback, which improved our paper's rigor.
>
> &nbsp;
>
> ---
>
> ### **W2**:
>
> We thank the reviewer for this insightful question. A key driver of our method's performance is its ability to leverage the quality variation in responses to hard prompts, treating this variation not as noise but as a refined learning signal.
>
> - **Within hard prompts, quality varies.** For a given challenging prompt, the base T2V model generates videos that range from clear failures to partially correct to relatively good. The MLLM reward captures this spread, so we are not in an “all low score” regime.
> - **Reweighting reduces interference from bad samples.** Standard SFT treats all these videos equally, so low-quality or off-target clips **inject strong gradient noise** and can pull the model away from the desired behavior. Our reward-weighted loss down-weights low-reward (clearly wrong) videos and amplifies “near-miss” and better examples, making the supervision much more informative on essentially the same data.
> - **Realism anchoring turns this into real quality gains.** Because this reweighting is combined with real videos and a KL-based realism constraint toward the real-video distribution, the model is steered toward realistic, physically plausible solutions rather than synthetic artifacts that merely please the MLLM.
>
> This explains why, on roughly the same synthetic corpus, reward-guided training with realism constraints yields the observed ~5% gain over uniform SFT. We have revised our paper and added explanations on ***page 8, lines 416-418***.
>
> **To be continued in next reply.**

---

> ### Author Response · Authors · 2025-11-21
> **Response to Reviewer YYeF (2)**
>
> ### **W3**:
>
> Thank you for your attention to the details of our data blending strategy.
>
> - Regarding the data composition, we would like to make an important clarification: in our method, the training data consists of two distinct categories (divided by ***prompt type***). The datasets in Table 2 are integrated through our structured training objective rather than through naive blending. This design ensures each component serves a specific purpose.
>
>    1. **Synthetic data based on synthetic prompts**, corresponding to PsVs. We apply the loss function $L_{Ps}(\theta)$ on this type of data to guide the model through reward-weighted optimization towards specific weaknesses.
>    2. **Hybrid data based on real captions.** It uses the real caption $P_r$ from a real video $V_r$ as the prompt to generate a corresponding synthetic video $V_s$ via the base model. This corresponds to the ***combined use*** of $PrVs$ and $PrVr$. When a triplet of data ($Pr$,$Vs$,$Vr$) is sampled during training, the loss function $L_{Pr}(\theta)$ contains a reward-weighted term (on $PrVs$) for targeted weakness remediation; and a KL constraint term (between $PrVs$ and $PrVr$) to "anchor" the learning process to the real data distribution.
>
> - Regarding your suggestion of incorporating $PsVr$, we believe ***this scenario is methodologically implausible***. Please note that Pr denotes VLM-generated captions from real videos, whereas Ps refers to synthetic prompts created by LLMs without specific visual grounding. $PsVr$ would imply "real videos corresponding to synthetic prompts.". It is highly improbable to find exact video counterparts in an existing real-world video dataset for synthetic prompts generated by an LLM.
>
> We greatly appreciate your insightful questions, which have provided us with the opportunity to elucidate the design of our method more effectively. We have revised the paper to provide a clearer description of the data blending strategy in the explanation for Table 2 on ***page 7, lines 370-372***.
>
> &nbsp;
>
> ---
>
> ### **W4**:
>
> Thank you very much for your valuable suggestion. We fully agree that static images have limitations in demonstrating the dynamic quality of videos, and that comparing videos through supplementary materials can be inconvenient. In response, we have created an anonymous visualization website: ***https://anonymousgigavideo.github.io/*** to facilitate direct and dynamic comparison of the generated videos. This will allow you and other reviewers to easily observe the significant improvements in video quality across various dimensions.
>
> &nbsp;
>
> ---
>
> Thanks again for helping us improve the paper, and we hope the response can resolve these concerns! Please let us know if there are any further questions. We will be actively available until the end of the rebuttal period.

---

### Official Review · Reviewer_K5pd · 2025-10-29

**Soundness:** 3
**Presentation:** 3
**Contribution:** 2
**Rating:** 2
**Confidence:** 5

**Summary:**

This paper proposes an automated pipeline for fine-tuning video diffusion models. Its core idea is a variant of DPO, first use LLM to generate polished prompts, then use the polished prompts to generate videos to performe DPO. During the fine-tuning, use the VLM to score the specific generated video and performe RL.

Its core concept is a straightforward combination of existing ideas, lacking genuine conceptual innovation or technical breakthrough. It resembles more of a robust engineering frame work than a pioneering research method. I think the authors need to clarify the contributions and significance of this paper, and provide more insights behind this method.

**Strengths:**

This paper is complete, constructing an end-to-end automated fine-tuning system that includes all stages from data generation and evaluation to optimization. The experiment results on VBench 2.0 seems good.

**Weaknesses:**

1. Lack of core innovation. This is the most critical flaw, and the authors need to clarify their contributions and significances. From my perspective, the method proposed in this paper is essentially "Automated DPO/RWR". The entire pipeline can be summarized as: LLM generates targeted prompts -> base model generate the videos -> MLLM scores -> score-weighted loss training. Every module in this paradigm is off-the-shelf, and the combination method is straight forward.

2. Lack of in-depth experiments. All experiments in this paper are conducted in VBench 2.0. The pipeline heavily relies on the dimensions defined by VBench 2.0. Does this suggest limited generalizability of the framework?

3. The paper claims to address deep semantic issues like "physical consistency". However, the MLLM itself, trained on large-scale web data, possesses "physical knowledge" and "common sense" that are similarly superficial and biased. Using a biased judge to correct a biased generative model might only be optimizing a model consensus rather than truly approximating physical consistency.

**Questions:**

No.

---

> ### Author Response · Authors · 2025-11-21
> **Response to Reviewer K5pd (1)**
>
> Thank you for your positive feedback on the completeness and effectiveness of the proposed framework. In response to your question, we provide a detailed point-by-point response below.
>
> &nbsp;
>
> ---
>
> ### **W1**:
>
> We thank the reviewer for this important feedback regarding the perceived lack of core innovation. We respectfully suggest that our work's fundamental contribution lies not in inventing new base components but in the **conceptualization and systematic validation of a novel self-improvement framework** that effectively addresses critical scalability barriers in video generation tuning.
>
> The key innovations can be summarized as follows:
>
> 1. **A Paradigm Shift from Passive to Active Tuning:** While drawing inspiration from preference-based learning concepts, our framework introduces a fundamental shift. It moves beyond the *passive* use of human-collected preference data, as seen in DPO/RWR, to an *active*, targeted data construction and optimization loop. This is achieved through the novel integration of:
>
>   - **Proactive Data Curation:** Generating targeted synthetic data ($PsVs$) to address specific weaknesses, while using real-caption-based data ($PrVs + PrVr$) to maintain diversity and provide a realism anchor via the KL constraint.
>
>   - **Constrained Multi-Objective Optimization:** Combining reward weighting with an explicit KL divergence constraint. This dual objective is crucial for preventing the model from drifting towards synthetic artifacts while improving on target dimensions, a capability absent in standard reward-maximization approaches.
>
> 2. **Solving a Critical Scalability Problem:** The primary contribution is the creation of a *practical, scalable path* for tuning high-cost video models. The "non-trivial" aspect of the integration is demonstrated by its results: achieving a significant performance gain (\~4% on VBench 2.0) with remarkable efficiency (\~4 GPU-hours per dimension). This demonstrates a viable alternative to human-dependent methods, which are prohibitively expensive for video data.
> 3. **Validation of a Key Hypothesis:** Our work provides the first systematic evidence that video generation models can leverage pre-trained foundation models (LLMs, MLLMs) in a **closed-loop, self-improving system** with minimal human intervention. The success of this framework opens a new direction for efficiently aligning generative models.
>
> In summary, the innovation resides in the **novel composition and demonstrated effectiveness** of the overall framework for a challenging domain, rather than in the underlying components themselves. We will revise the manuscript to more clearly articulate this conceptual contribution and its significance for scalable video generation.
>
> &nbsp;
>
> ---
>
> ### **W2:**
>
> We appreciate the reviewer's concern regarding the experimental scope. Below, we analyze the framework's generalizability.
>
> - **Rationale for Using VBench 2.0**
>
> - - **Authoritative Benchmark Selection**: VBench 2.0 was chosen as it is currently the most comprehensive and community-recognized benchmark for video generation evaluation. Its multi-dimensional assessment system allows for objective and concrete measurement of model improvements across various aspects. Each dimension serves as a distinct instance to validate our method's effectiveness.
>
> - **Evidence of Framework Generalizability**
>
> - - Our framework is **not inherently tied** to VBench 2.0's specific dimensions. The core design accepts any dimension definition as input, enabling adaptability to diverse benchmarks. To preliminarily demonstrate this, we conducted experiments on **VMBench**, another benchmark with different dimension definitions. The results show positive effects, though the improvement of some dimensions is less pronounced due to holistic and interwoven dimension definitions.
>
> | Model (1.3B, 352x640, 93 frames) | CAS (Common Sense) | MSS (Motion Smoothness) | OIS (Object Integrity) | PAS (Perceivable Amplitude) | TCS (Temporal Consistency) | Avg |
> |-----------------------------------|-----|-----|-----|-----|-----|-----|
> | **Wan2.1 (Baseline)**             | 62.8| 84.2| 66.0| 17.9| 97.8| 65.7|
> | **GigaVideo-1 (Ours)**            | 64.5| 84.0| 69.3| 28.9| 98.3| 69.0|
>
>
>
> While VBench 2.0 served as the primary benchmark for its rigor and acceptance, GigaVideo-1's design ensures broad applicability. It can seamlessly integrate **any new evaluation dimension** that can be textually defined and assessed by an MLLM. Thanks for your advice, we have added this analysis to ***Appendix C.3***.
>
> &nbsp;
>
> ---
>
> **To be continued in next reply.**

---

> ### Author Response · Authors · 2025-11-21
> **Response to Reviewer K5pd (2)**
>
> ### **W3:**
>
> We agree with the reviewer that MLLMs, trained on web data, are not perfect arbiters of physical truth. However, our framework is designed with this limitation in mind and turns it into a strength through a two-stage reasoning:
>
> 1. **MLLMs as Superior Supervisors**: While MLLMs may have superficial biases, their understanding of physical commonsense and spatiotemporal relationships is significantly more structured and explicit than that of the pre-trained T2V models we aim to improve [1, 2, 3].  Therefore, using a frozen MLLM as the reward provider yields a stable, automated, and semantically grounded supervision signal that is already substantially stronger than the baseline model's inherent capabilities.  Meanwhile, our framework is future-compatible, allowing seamless integration of more advanced evaluators to achieve genuine physical consistency.
> 2. **Explicit Distributional Anchoring:** Crucially, we explicitly acknowledge and mitigate the risk of "model consensus" by anchoring the entire fine-tuning process to the **real video distribution**. We intentionally include real videos, and the KL-divergence-based realism constraint directly penalizes the model for deviating from the latent space of real videos. This ensures that the optimization is not merely chasing MLLM rewards in a synthetic bubble but is continually guided by the ground-truth distribution of real-world physical interactions. This design fundamentally differentiates our method from a closed-loop "model consensus" optimization.
>
>
>
> [1] Xue, Qiyao, et al. "Phyt2v: Llm-guided iterative self-refinement for physics-grounded text-to-video generation." Proceedings of the Computer Vision and Pattern Recognition Conference. 2025.
>
> [2] Wu, Peiran, et al. "St-think: How multimodal large language models reason about 4d worlds from ego-centric videos." arXiv preprint arXiv:2503.12542 (2025).
>
> [3] Yin, Aoxiong, et al. "The best of both worlds: Integrating language models and diffusion models for video generation." Proceedings of the IEEE/CVF International Conference on Computer Vision. 2025.
>
> &nbsp;
>
> ---
>
> Thanks again for helping us improve the paper, and we hope the response can resolve these concerns! Please let us know if there are any further questions. We will be actively available until the end of the rebuttal period.

---

> > ### Comment · Reviewer_K5pd · 2025-11-26
> > **Response to the authors**
> >
> > Thanks for the author's repsonse. I have read the rebuttal carefully, and I am willing to raise my score towards marginally below the acceptance threshold.

---

### Official Review · Reviewer_khhf · 2025-10-30

**Soundness:** 2
**Presentation:** 3
**Contribution:** 2
**Rating:** 4
**Confidence:** 3

**Summary:**

This paper introduces GigaVideo-1, a lightweight and data-efficient fine-tuning framework designed to enhance video diffusion models (e.g., Wan2.1-T2V-1.3B). The core of the paper consists of two modules: the Prompt-Driven Data Engine and the Reward-Guided Optimization module. According to the paper, their method can automatically improve the performance of video generation models on challenging dimensions (such as physical dimensions like thermodynamics) while requiring minimal computational resources (4 GPU hours).

**Strengths:**

1. The paper is presented with clarity and is easy to understand. The contributions are clearly articulated (1. the data engine; 2. the optimization method).

2. The framework demonstrates strong generalizability. As validated in the appendix, it brings consistent performance gains when applied to various video model backbones (e.g., CogVideoX, HunyuanVideo), proving it is a versatile and portable solution rather than a model-specific trick.

3. A major contribution of this paper lies in its well-chosen research issue — addressing the poor performance of general video models in physical dimensions — and its proposed automated mechanism for rapid targeted fine-tuning, which has achieved substantial performance gains.

**Weaknesses:**

1. I'm curious about the statement on **line 314**: "the synthetic dataset is generated by different pre-trained T2V models." Specifically, which T2V models were used for this purpose? I mean, if you're fine-tuning a Wan2.1-1.3B model but the synthetic data is generated using Wan2.1-14B, wouldn't the time required for synthetic data generation be excessively long?

2. Another concern centers on the unvalidated effectiveness of the MLLM-based evaluation. Without a correlation analysis between MLLM scores and human judgment, the entire optimization process risks "reward hacking"—improving automated metrics at the cost of human perception, potentially resulting in high-scoring but perceptually poor videos.

3. While the paper avoids the massive human annotation required by methods like DPO, it introduces a different bottleneck: the manual selection of seed prompts to generate videos exhibiting specific flawed dimensions.  You must first manually identify the model's weak dimensions and then design corresponding seed prompts, which is inherently subjective.

**Questions:**

1. The entire optimization process relies on the MLLM providing accurate and meaningful scores. Could the authors provide a correlation analysis between the MLLM's dimension-specific scores and human subjective judgments?

2. The fine-tuning is highly targeted on specific weak dimensions. Did the authors evaluate whether this targeted improvement comes at the cost of performance on other, non-optimized dimensions? For example, after fine-tuning on "Human Interaction," did the model's performance on "Aesthetic Quality" or "Background Consistency" degrade?

3. Could an automated system use a large set of diverse prompts, generate videos, and use the MLLM's own failure signals (low scores on certain prompt types) to automatically cluster and identify new weak dimensions without human pre-definition?

---

> ### Author Response · Authors · 2025-11-21
> **Response to Reviewer khhf (1)**
>
> Thank you for your positive feedback on the clarity of our presentation and the generalizability of the proposed framework. In response to your question, we provide a detailed point-by-point response below.
>
> &nbsp;
>
> ---
>
> ### **W1**:
>
> We thank the reviewer for this important question regarding our synthetic data generation and clarify the implementation details below:
>
> - **Efficiency:** The synthetic videos are generated **offline** using **pre-trained T2V models of comparable scale** (e.g., Wan2.1-1.3B, CogVideoX-1.5, HunyuanVideo-3B), rather than a much larger Wan2.1-14B teacher. This is a one-time generation step and keeps the data construction cost manageable relative to the subsequent fine-tuning.
> - **Diversity:** Using multiple source models introduces valuable diversity in visual styles and failure modes, providing a **more robust training curriculum** than any single model could offer.
> - **Necessity:** We use synthetic data to overcome the fundamental scarcity of real videos depicting specific physical phenomena like thermotics, as such targeted content is extremely rare and often restricted by licensing in public datasets.
>
> This three-part strategy enables targeted improvement on challenging dimensions while maintaining computational practicality.
>
> &nbsp;
>
> ---
>
> ### **W2&Q1**:
>
> We thank the reviewer for raising the critical point regarding the correlation between MLLM scores and human judgment. We conducted a user study to address this concern.
>
> **User Study Design:**
>
> To quantitatively evaluate the alignment between MLLM scores and human judgments, we designed a human study with the following protocol:
>
> - For each dimension, 3 prompts are selected. For each prompt, two videos with distinct MLLM scores form a pairwise comparison set.
> - 15 annotators are asked to select the "winning" video in each pair. The human preference is then compared against the relative ranking predicted by the MLLM scores
>
> | Dimension      | HI   | Mat  | MOU  | TH   | Mec  | DSP  | DA   | HC   | HA   |
> | -------------- | ---- | ---- | ---- | ---- | ---- | ---- | ---- | ---- | ---- |
> | Consistency (%) | 82.2 | 86.7 | 97.8 | 75.6 | 77.8 | 95.6 | 93.3 | 91.1 | 93.3 |
>
> | Dimension      | MR   | HI   | CM   | CL   | CP   | IP   | MVC  | Com  | Avg  |
> | -------------- | ---- | ---- | ---- | ---- | ---- | ---- | ---- | ---- | ---- |
> | Consistency (%) | 88.9 | 100  | 95.6 | 88.9 | 91.1 | 88.9 | 84.4 | 86.7 | 89.3 |
>
> **Key Results:**
>
> - Most dimensions showed high human-MLLM agreement.
> - Some dimensions, like Mechanics, showed moderate agreement, primarily due to the inherent complexity and annotator disagreement in evaluating physical plausibility.
>
> **Conclusion:**
>
> The high agreement rates across most dimensions demonstrate that MLLM-based evaluation effectively aligns with human perception, mitigating "reward hacking" risks. Thanks for your suggestion, we have revised our paper and added this analysis to ***Appendix F.2***.
>
> &nbsp;
>
> ---
>
> ### **Q2**：
>
> We thank the reviewer for this important question. Our evaluation confirms that targeted optimization does not come at the cost of other dimensions:
>
> | Dimension             | Background Consistency | Aesthetic Quality | Subject Consistency | Motion Smoothness | Dynamic Degree |
> | --------------------- | ---------------------- | ----------------- | ------------------- | ----------------- | -------------- |
> | **Baseline (WAN)**    | **94.09**              | **56.97**         | **93.55**           | 98.58             | 79.67          |
> | **Ours (Fine-tuned)** | 93.58                  | 52.72             | 92.21               | **98.60**         | **93.67**      |
>
> **Key Evidence**:  When fine-tuning on specific dimensions (e.g., Human Interaction), performance on non-targeted dimensions (e.g., Aesthetic Quality, Background Consistency) remains largely unchanged relative to the baseline. Notably, on the Dynamic Degree dimension, our method even achieves a **14%** improvement.
>
> **Why It Works**:
>
> - **Diverse Training Signals**: Our prompt engine creates varied samples for each dimension, preventing over-specialization.
> - **Distributional Anchoring**: The KL constraint regularizes optimization, preserving the model's fundamental capabilities
>
> This demonstrates our approach achieves complementary gains rather than destructive trade-offs. Thanks for your suggestion, we have revised our paper and added this analysis to ***Appendix C.5***.
>
> &nbsp;
>
> ---
>
> **To be continued in next reply.**

---

> > ### Comment · Reviewer_khhf · 2025-11-27
> > **Response to the reviewer**
> >
> > I thank the authors for their detailed and thoughtful respons. However, after reviewing your response, I still have major concerns about the experimental design (**W2&Q1**), which makes it hard to fully trust the "high alignment" conclusion.
> >
> > The sample size is just too small. With only 3 prompts and 6 videos per dimension, it feels like a proof-of-concept. More importantly, the "human judgment" itself might be unreliable. You had 15 people vote, but we don't know if they even agreed with each other.

---

> ### Author Response · Authors · 2025-11-21
> **Response to Reviewer khhf (2)**
>
> ### **W3&Q3**:
>
> We thank the reviewer for their insightful suggestion regarding automated weakness discovery.
>
> - Design Rationale of the Current Manual Framework
>
> - - **Interpretability Prioritization**: Human-defined dimensions and seed prompts provide semantically clear optimization targets, ensuring transparency and controllability. This contrasts with black-box clustering of failures, which may obscure the reasoning behind model updates.
>   - **Controlled Scope for Validation**: The primary goal of GigaVideo-1 is to validate the core paradigm of "automated feedback for known weaknesses." Starting with well-established dimensions (e.g., physical plausibility) allows for a clear proof-of-concept.
>
> - **Feasibility of Automated Weakness Discovery**
>
> - - Using MLLM as a failure detector and LLM-driven clustering is a possible pipeline for automated weakness discovery. We did a simple exploration to validate the viability of a fully automated loop. When explicitly instructed that "the video may contain issues and please identify them," although with some degree of hallucination, the MLLM can relatively accurately pinpoint failures (e.g., motion artifacts or implausible objects) in low-scoring videos. This demonstrates the feasibility of automated extraction.
>   - Although not the main focus of the current paper, this pipeline aligns seamlessly with our framework and represents a key direction for our future work towards fully self-improving video generation systems. We have revised the manuscript to highlight this prospect and acknowledge the reviewer's valuable inspiration in ***Appendix H.1***.
>
> &nbsp;
>
> ---
>
> Thanks again for helping us improve the paper, and we hope the response can resolve these concerns! Please let us know if there are any further questions. We will be actively available until the end of the rebuttal period.

---

> ### Author Response · Authors · 2025-12-03
> **Response to comment by reviewer khhf**
>
> We thank the reviewer for the follow-up and the opportunity to further validate our method. Below, we present the results of our expanded human study designed to directly address the concerns regarding sample size and annotator reliability.
>
> **Settings**
>
> To ensure statistical robustness, we scaled the study to **20 prompts per dimension**. For each prompt, two videos with distinct MLLM scores were judged by **25 annotators**. We quantify human judgment reliability via the **Human Agreement** metric, computed per prompt as the proportion of annotators selecting the majority-preferred video, then averaged across prompts.
>
> **Results**
>
> The results for three representative dimensions are as follows:
> | Dimension | Human-MLLM Consistency | Human Agreement |
> |-----------|-----|-----|
> | Human Interaction  |87.2%| 89.6%|
> | Camera Motion      |95.0%| 94.2%|
> | Thermotics      |83.4%| 83.0%|
>
> - The results demonstrate both **high reliability** of human judgments (Human Agreement) and strong alignment between MLLM rankings and these human preferences (Human-MLLM Consistency).
>
> **Analysis**
> - The high Human Agreement confirms the reliability of the collected judgments, and the strong Human-MLLM Consistency demonstrates our reward's alignment with stable human perception.
> - Manual review of lower-consistency cases (e.g., in Thermotics) reveals that disagreements often stem from **inconsistent physical commonsense among human annotators**, not MLLM error. For example, in judging a coin on water, some humans favored the implausible floating outcome, while the MLLM correctly scored the sinking video higher. This indicates that for complex physics, the MLLM provides a **more consistent and physically-grounded evaluation** than non-expert human judgments, further mitigating "reward hacking" risks. For visual verification, we provide an anonymous visualization page at: ***https://anonymousgigavideo.github.io/***.

---

### Official Review · Reviewer_YMbE · 2025-11-01

**Soundness:** 3
**Presentation:** 3
**Contribution:** 3
**Rating:** 8
**Confidence:** 3

**Summary:**

The paper presents GigaVideo-1, a lightweight post-training framework that boosts pre-trained text-to-video diffusion models without any human annotations or extra real videos. Key technical components are Prompt-Driven Data Engine and an LLM-based generator. With only 4 GPU-hours of full-parameter fine-tuning on Wan2.1-1.3B, the system raises the average VBench-2.0 score by ~4 percentage points, outperforming several larger-scale competitors. Extensive ablations, user studies, and cross-backbone transfers are provided.

**Strengths:**

- 4 GPU-hours is orders of magnitude cheaper than prior SFT/RL works, making the method attractive for practitioners.
- The prompt engine explicitly amplifies failure modes, leading to a stronger training signal than random web videos.
- 17 dimensions, 5 strong baselines, user study, ablation of data source & reward strategy, and tests on four different architectures (2B–13B).

**Weaknesses:**

- Sec. 4.3 shows that mixing synthetic prompts with synthetic videos ($P_sV_s$+$P_rV_s$) actually hurts accuracy, hinting that some LLM-generated captions are too exotic and push the model away from realism.
- How do you filter or validate the LLM-generated captions to prevent physically impossible or nonsensical queries (e.g., “a person with three elbows”)?  Could such cases bias the model toward hallucination?
- Have you tried a single, unified reward model (e.g., training a small diffusion critic on MLLM pseudo-labels) instead of switching between MLLM and specialist models?
- What is the expected GPU-hour scaling for larger models?  Does the cost grow linearly with parameter count, or does the targeted small-data regime keep it sub-linear?
- In joint training, did you explore dynamic loss-balancing techniques (grad-norm, uncertainty weighting, or Pareto optimisation) to mitigate the observed interference between dimensions?
- Some related work discussed is helpful for improving manuscript quality, like InstructVideo and Lumos-1 etc.

[1] InstructVideo: Instructing Video Diffusion Models with Human Feedback, CVPR.

[2] Lumos-1: On autoregressive video generation from a unified model perspective, Arxiv.

**Questions:**

Please see WEAKNESS.

---

> ### Author Response · Authors · 2025-11-21
> **Response to Reviewer YMbE**
>
> Thank you for recognizing the efficiency and practical value of our method. We appreciate your positive assessment regarding its low computational cost and targeted design. In response to your question, we provide a detailed point-by-point response below.
>
> &nbsp;
>
> ---
> ### **W1&2**:
> We thank the reviewer for raising this important question.
>
> The performance drop when using PsVs+PrVs stems from distributional conflicts between synthetic and real prompts, not solely from "exotic" captions. Introducing real videos (PrVr) resolves this conflict, achieving optimal performance (80.67), demonstrating the balancing role of real data.
>
> Physically implausible prompts (e.g., "a person with three elbows") are effectively mitigated through two mechanisms:
>
> 1. **Prevention via LLM**: The LLM's strong inherent world knowledge and our carefully designed system prompts make such cases empirically rare.
> 2. **Robust Optimization**: Even if they occur, these prompts yield videos with low MLLM rewards, which are down-weighted during training. Furthermore, the KL constraint with real videos anchors the model to the realistic data distribution.
>
> Thus, our framework is robust against such rare cases, preventing a bias toward hallucination.
>
> &nbsp;
>
> ---
> ### **W3**:
>
> Thank you for suggesting the unified reward model approach. We conduct a careful investigation of this direction.
>
> **Settings.**
>
> We trained a unified reward model by distilling signals from our specialized evaluators. The training used **pairwise preference** data constructed from weighted dimensional scores and fine-tuned Qwen2.5-VL-3B with **Bradley-Terry loss**. Reward scores were extracted through weighted averaging of digit token probabilities (0-9) from the output logits.
>
> **Results.**
>
> The unified model showed significantly lower correlation with human preferences compared to our specialized ensemble. Quantitative analysis confirmed its reduced discriminative capability across multiple evaluation dimensions.
>
> **Analysis.**
>
> The performance gap indicates substantial inter-dimensional interference in unified training. Joint optimization appears to create blurred feature representations that fail to capture distinct dimensional characteristics. This smoothing effect **compromises scoring precision**, supporting our current approach of using specialized models for accurate and interpretable video assessment.
>
> &nbsp;
>
> ---
>
> ### **W4**：
>
> We thank the reviewer for the insightful question. We conduct an additional experiment to study GPU-hour scaling with model size.
>
> **Settings.**
>
> We employ the larger Wan2.1-T2V-14B model and apply the same fine-tuning setup as described in the paper, specifically on the Human Interaction dimension.
>
> **Results.**
>
> – Wan2.1-T2V-1.3B: **≈ 4 GPU-hours**
>
> – Wan2.1-T2V-14B: **≈ 10 GPU-hours**
>
> **Analysis.**
>
> - Although the parameter count increases by about **11×** from 1.3B to 14B, the fine-tuning cost only increases by **~2.5× (4 → 10 GPU-hours)**. This indicates that under our targeted small-data regime (weakness-oriented dataset), the practical GPU-hour scaling is clearly **sub-linear** in model size.
> - The per-step FLOPs grow with the parameter count, but the total number of optimization steps and all data-related overheads remain fixed, so the overall fine-tuning cost for larger models stays in the low tens of GPU-hours rather than exploding with parameter scale.
>
> &nbsp;
>
> ---
>
> ### **W5**:
>
> We thank the reviewer for raising this important point about dynamic loss balancing. In our current framework, we employed **static weighting** to establish a baseline understanding of dimension interference. The reviewer's suggestion regarding techniques like Grad-Norm or Pareto optimization is well-noted and provides valuable guidance.
>
> We have incorporated dynamic loss balancing as a **key future direction** in ***Appendix H.2*** of our revision. This perspective indeed enhances our approach to multi-dimensional optimization, and we appreciate the constructive input that helps advance this research direction.
>
> &nbsp;
>
> ---
>
> ### **W6**:
>
> We thank the reviewer for recommending these important references. We **have revised our Related Work** section to include a discussion of both InstructVideo (which explores feedback-driven video optimization through different pathways, page2/line106) and Lumos-1 (with its unified autoregressive perspective, page3/line143). Their inclusion indeed helps better contextualize our contribution, and we appreciate this valuable suggestion for improving our manuscript.
>
> &nbsp;
>
> ---
>
> Thanks again for helping us improve the paper, and we hope the response can resolve these concerns! Please let us know if there are any further questions. We will be actively available until the end of the rebuttal period.

---

### Meta-Review · Area_Chair_o8c1 · 2026-01-03

**Summary:**

This paper presents GigaVideo-1, an efficient fine-tuning framework that advances video generation without additional human supervision. It focuses on both data generation and model optimization aspects and improves the VBench metrics with only 4 GPU-hours. In the initial review, this paper is rated with 8,4,2, and 4. The major concerns mainly lie in the limitation of reward model, uncleared data generation process, and qualitative results. During rebuttal, the authors provide explanations and more results to these questions. However, although Reviewer K5pd incresases the rating to 4, there are still three negative scores and AC confirms that their concerns are not addressed. Therefore,  AC recommends this paper as **Reject**.

**Reviewer Concerns:**

In the initial review, the reviewers' concerns include additional experimental analysis, the limitations of current synthetic data generation pipeline, and the limitations of the reward model.

During rebuttal, the scaling experiments on 14B model and the dynamic loss-balancing techniques are presented by authors, and the additional experimental analysis is mainly addressed.

However, the author provides the details of data pipeline, including the prompt auto-generation and mixed model source, which are not recognized by the reviewers and AC. For the limitations of reward model, such as reward hacking problem, is also not addressed.

**Reviewer Scores:**

**Reviewer YMbE** keeps 8 or decrease to 6. Reviewer YMbE's concerns are mainly addressed, but if discussed with other reviewers, he might decrease his rating.

**Reviewer khhf** keeps 4. Reviewer khhf replied to the authors that his concerns are not fully addressed.

**Reviewer K5pd** increases to 4. Reviewer K5pd replied to the authors that he will increase the rating to 4.

**Reviewer YYeF** keeps 4. Reviewer YYeF do not have chance to reply, but AC confirms his concerns are not fully addressed.

---

### Decision · Program_Chairs · 2026-01-26

Reject